# Exploring the Diversity of Microbial Communities Associated with Two *Anopheles* Species During Dry Season in an Indigenous Community from the Colombian Amazon

**DOI:** 10.3390/insects16030269

**Published:** 2025-03-04

**Authors:** Daniela Duque-Granda, Rafael José Vivero-Gómez, Laura Alejandra González Ceballos, Howard Junca, Santiago R. Duque, María Camila Aroca Aguilera, Alejandro Castañeda-Espinosa, Gloria Cadavid-Restrepo, Giovan F. Gómez, Claudia Ximena Moreno-Herrera

**Affiliations:** 1Grupo de Microbiodiversidad y Bioprospección, Laboratorio de Biología Celular y Molecular, Laboratorio de Procesos Moleculares, Facultad de Ciencias, Universidad Nacional de Colombia at Medellín, Street 59A #63-20, Medellín 050003, Colombia; daduquegr@unal.edu.co (D.D.-G.); lgonzalezce@unal.edu.co (L.A.G.C.); acastanedae@unal.edu.co (A.C.-E.); gecadavi@unal.edu.co (G.C.-R.); 2RG Microbial Ecology: Metabolism, Genomics & Evolution, Div. Ecogenomics & Holobionts, Microbiomas Foundation, LT11A, Chia 250008, Colombia; howard.junca@gmail.com; 3Grupo de Limnología Amazónica, Universidad Nacional de Colombia at Amazonía, Kilómetro 2, Vía Tarapacá, Leticia 910001, Colombia; srduquee@unal.edu.co (S.R.D.); marocaa@unal.edu.co (M.C.A.A.); 4Grupo de Artropodología Básica y Aplicada, Universidad Nacional de Colombia at La Paz, Kilómetro 9, Vía Valledupar, La Paz 202010, Colombia; gfgomezg@unal.edu.co

**Keywords:** microbiota, *Anopheles darlingi*, *Anopheles triannulatus* s.l., endosymbionts, Amazon region

## Abstract

The Amazon Basin reports the highest number of malaria cases in the Americas, with control efforts hampered by parasite drug resistance and insecticide resistance in *Anopheles* mosquitoes, the insect vectors of this disease. As ongoing environmental changes in the Amazon may further influence malaria spread, innovative control strategies are essential. Bacterial communities within *Anopheles* species offer promise for biological control, as some bacteria have antiparasitic potential and can alter mosquito immune and reproductive systems, disrupting malaria transmission. For this reason, this study aimed to explore the bacterial communities in two *Anopheles* species collected from Leticia, Amazonas, Colombia, during the dry season. DNA extraction and sequencing revealed differences in the bacterial communities between species. Certain bacterial genera were associated with the inhibition of parasite development, while others appeared beneficial for mosquito development, as reported in the literature. These findings highlight the potential of these microorganisms as a tool for biotechnological interventions, although the specific roles of these bacteria in the analyzed species still require further investigation.

## 1. Introduction

The *Anopheles* (*An.*) genus includes species of public health importance because they are responsible for transmitting protozoan parasites of the genus *Plasmodium* (*P.*), which are the causative agents of malaria, filarial worms like *Wucheriera bancrofti*, *Brugia malayi*, and *B. timori*, and viruses such as o’nyong-nyong (ONNV) and Venezuelan equine encephalitis virus (VEEV) [1,2,3,4,5] among others [6].

Since 2015, malaria cases around the world are rising, with the largest increase in estimated cases and deaths occurring between 2019 and 2020 [7]. In the Americas, 73% of cases are reported in Venezuela, Colombia, and Brazil, and more than 90% of the regional malaria burden is associated with the Amazon Basin, with *P. vivax* being the predominant parasite [7,8]. In the Colombian Amazon, the main malaria vectors are *An. darlingi* and *An. benarrochi* B, while other species such as *An. triannulatus* s.l. and *An. punctimacula* are considered local vectors [9,10,11,12], and in Leticia, a malaria-endemic town in the Amazon with both rural and urban settlements, there have been 414 of cases reported in 2024 (9 coming from the San Pedro de los Lagos locality), with reports of *Plasmodium* infections associated with *P. vivax*, *P. malariae*, and *P. falciparum*, with cases of mixed infections [13,14,15].

The vector competence of *Anopheles* mosquitoes can be affected by a tripartite interaction that occurs among the parasite, the mosquito, and its microbiota, as *Plasmodium* protozoans go through the gut and salivary glands of the insect that are already colonized by microorganisms impacting the establishment of the parasite, thus affecting its transmission [16]. These microorganisms can be acquired through vertical transmission or by the exposure of larvae and pupae at the breeding sites remaining at later life stages through transstadial transmission [17]. The microorganisms present in mosquito breeding waters can impact egg-laying and larval development, change the bacterial composition in the guts of adult mosquitoes, and, in turn, influence the transmission of human pathogens, such as malaria parasites [18].

Similarly, abiotic factors influence vector competence as aquatic conditions in larval breeding sites can affect mosquito development, fitness, and vectorial capacity [19]. In this regard, researchers have evaluated the microbiota at breeding sites and its physicochemical characteristics in Kenya to further understand larval ecology [20], finding that the most common phyla were Pseudomonadota, followed by Bacteroidota and Actinomycetota and that there was a negative correlation between the presence of *Anopheles* mosquitoes and parameters such as electrical conductivity, salinity, ammonia, and total dissolved solids. Other authors [19] found that the core bacterial communities across breeding sites for different mosquitoes (*Anopheles*, *Aedes*, and *Culex*) were *Flavobacterium*, *Dechloromonas*, *Mycobacterium*, *Hydrogenophaga*, *Clostridium*, and *Rubrivivax*, and that *Anopheles* mosquitoes were positively correlated with dissolved oxygen, nitrates, and temperature. It has also been suggested that other environmental variables such as seasonal changes may have an impact in insect gut microbiota, where findings show significant differences in bacterial composition during wet and dry seasons potentially influencing mosquito biology [21].

Microbiota in different *Anopheles* species evidence the presence of bacteria of the genera *Thorsiella*, *Pantoea*, *Pseudomonas*, *Serratia*, and *Asaia* in *An. atroparvus* [22], with the last two being identified as well in *An. gambiae* along with bacteria of the genera *Elizabethkingia* and *Wigglesworthia* [23]. Most studies in the Americas are focused on the Brazilian Amazon basin, and there is a lack of information regarding microbiota–*Anopheles* interactions in the Colombian Amazon. In this regard, studies of communities living close to the Brazilian Amazon river 1000 km downstream (east) of the Colombian frontier evidence the presence of Peptostreptococcaceae, Neisseriaceae, Oxalobacteraceae, and Streptococcaceae families in immature individuals of *An. darlingi*, as well as Enterobacteriaceae, Bacillaceae, Pseudomonadaceae, and Staphylococcaceae families at breeding sites [24].

Other authors specifically identified *Escherichia/Shigella*, *Staphylococcus*, and *Pseudomonas* in *An. darlingi* breeding waters [18], and *Acinetobacter*, *Enterobacter*, *Klebsiella*, *Serratia, Elizabethkingia*, *Stenotrophomonas*, *Bacillus*, and *Pantoea* bacteria in the same species of mosquito and water bodies, identifying *Pantoea* and *Serratia* as possible candidates for paratransgenesis strategies [25]. In other species, such as *An. albimanus* from Colombia, *Bacillus, Enterobacter*, and *Staphylococcus* have been identified as the core bacteria of this mosquito, while in *An. nuneztovari* and *An. darlingi* from the Colombian Pacific and Caribbean coasts Enterobacteriaceae, Comamonadaceae, Aeromonadaceae, Pseudomonadaceae, Moraxellaceae, and Rhodocyclaceae are among the most predominant families in adults [26,27].

Current mosquito control methods encompass the drainage and weeding of potential breeding sites, the application of chemical insecticides such as pyrethroids, and biological control measures like the use of natural predators and *Bacillus thuringiensis israelensis* (Bti) toxins for larval control [28]. However, these strategies frequently fail due to high maintenance costs, the development of resistance to larvicides or insecticides, conditional effectiveness specific to Bti, or insufficient evidence to evaluate the success of interventions involving larvivorous fish [28,29]. Consequently, there is a pressing need to develop new biological control strategies that consider the interactions between mosquito vectors and their microbiota and endosymbionts, as recent studies indicate that endosymbiotic bacteria significantly influence insect biology by affecting pathogen development through metabolite production or stimulation of the host immune system [30].

There is evidence that endosymbionts have an impact on the development of *Plasmodium* parasites. For instance, *Wolbachia*, with prevalences varying from 1.21% to 57.1% [31,32,33,34,35,36], exhibits an inhibitory effect in the development of *P. falciparum* in *An. stephensi* by inducing the expression of immune-related genes such as CLIP domain serine proteases, lysozyme C2, PGRP-LD, and TEPs [36]. Similarly, *Microsporidian* endosymbionts, with prevalences from 1.8% to 57.0% in *Anopheles* populations [37,38,39,40], have been linked to hindering the full development of *Plasmodium* by preventing the parasite from reaching the salivary glands of the insect [37]. Other bacteria, such as *Spiroplasma,* have also been detected in *Anopheles* mosquitoes but at very low prevalences [41,42,43]. However, the roles of microbiota in anophelines from the Neotropics, particularly the Amazon Basin, and their impact on the malaria transmission cycle remain unclear. Given the limited information on the microbiota composition of these insect vectors in the region, especially in Colombia, it is essential to first identify the bacterial communities and those with biotechnological potential present in these insects.

Therefore, this study aimed to characterize the bacterial composition and to determine the presence of endosymbionts of biotechnological interest in the breeding sites and at different life stages of *An. triannulatus* s.l., local malaria vector, as well as in adults of *An. darlingi*, collected in the intradomestic and peridomestic areas of the San Pedro de los Lagos locality, in the municipality of Leticia, department of Amazonas, Colombia, which is located near the triple border of Brazil/Peru/Colombia, where the malaria burden is shared by the different countries.

## 2. Materials and Methods

### 2.1. Ethics Statement

*Anopheles* specimen collection was conducted following the parameters of Colombian Decree No. 1376 of 2013 of the Ministerio de Ambiente y Desarrollo Sostenible, and the Framework Permit for Collecting Specimens of Wild Species of Biological Diversity for Non-commercial Scientific Research Purposes, conceded to the Universidad Nacional de Colombia by the Resolution 0255 of 12 March 2014, of the Autoridad Nacional de Licencias Ambientales. Mosquitoes were collected in the locality of San Pedro de los Lagos (S.P.L), municipality of Leticia, department of Amazonas, and consent was received from the indigenous community before sampling.

### 2.2. Study Area and Collection of Samples

Samples were collected in the department of Amazonas, Colombia, in the municipality of Leticia, S.P.L. locality (Figure 1), in the Yahuarcaca watershed, tributary to the Amazon River. The locality is home to the Ticuna people, in a settlement with 102 inhabitants, at approximately 6.5 km from the urban area of Leticia (4°08′32.3″ S 69°57′13.8″ W), surrounded by crops of cassava, sugar cane, herbal medicines, and fruit and palm trees, as well as Amazonian secondary forest, with presence of domestic animals such as dogs, cats, and chickens without presence of livestock.

*Anopheles* specimens were collected in the locality of S.P.L. in Amazonian secondary forest during November of 2023, and in the intradomestic and peridomestic area of the same locality in January of 2024 (Figure 1). Temperature records at the site showed a mean value of 30 °C and a relative humidity above 90% during the sampling period [44]. Shannon and CDC light traps for collection of adult individuals were installed in Amazonian secondary forest (Figure 2) during a period of extreme warm conditions and low precipitations [45], while human landing catches (HLC) were used for the collection of adult mosquitoes with an entomological aspirator in the intradomestic and peridomestic area.

Active search of potential breeding sites was conducted in the vicinity for water sampling and collection of *Anopheles* larvae and pupae. The area corresponds to the Yahuarcaca stream watershed, where water samples were taken from palm tree bracts, seed husks, abandoned flooded canoes at water bodies, abandoned fishing ponds with no presence of fish, and puddles adjacent to the Yahuarcaca stream (Figure 2).

Longmire’s solution (100 mM Tris, 100 mM EDTA, 10 mM NaCl, 0.5% SDS, 0.2% sodium azide) was added to the water sample for preservation before processing for DNA extraction, following the protocol described by other authors [46]. Characterization of physicochemical properties of positive and negative breeding sites was conducted by sampling water for chemical oxygen demand (COD), total iron, nitrate, copper, and fixed and volatile solids.

Larvae and pupae collected from the breeding site were preserved as follows: Large sterile pipettes and larval dippers were used to gather available immature anophelines at the breeding site. Water from the site, along with immature individuals, was placed in a sterile flask covered with a muslin fabric to prevent the escape of newly emerged adults. Larvae were categorized by life stage (instar 1–2 or instar 3–4) and stored in sterile 1.5 mL vials containing ethanol (70%) for molecular analyses. A subset of pupae was immediately stored under the same conditions, while the remaining pupae were allowed to develop into adults to obtain representatives of all life cycle stages (except eggs). All samples were stored at −20 °C until whole-body DNA extraction was performed (Section 2.3).

Adult and immature *Anopheles* specimens were morphologically identified using the morphological keys of [47,48,49,50]. Then, morphotype representatives were selected for molecular species confirmation (see Section 2.4).

A total of 38 individuals were processed for sequencing using the Illumina Miseq platform (see Section 2.5), representing the life stages of *An. triannulatus* s.l and *An. darlingi* adults. The specimens were grouped into pools of 5 individuals each, except for 3 adults that emerged from individuals collected at the fishing pond, which were grouped to form the *An. triannulatus* s.l adult pool.

### 2.3. DNA Extraction of Immature Individuals, Adult Mosquitoes, and Water Sample from the Breeding Site

Before DNA extraction, the individuals were washed sequentially using Tween 20, sterile water, 1X Phosphate Buffer Saline (PBS) solution, and finally, ethanol (70%), with slow centrifugation at 3000 rpm for 2 min at each step. The ethanol was then removed and the whole bodies of *Anopheles* were dried before proceeding with DNA extraction.

To analyze the bacterial composition from breeding sites, immature individuals, and adult mosquitoes, DNA isolation was conducted as follows: The water sample was processed using the DNeasy PowerWater Kit (Qiagen^®^, Hilden, Germany) after sediment removal through filtration with a 0.20 µm filter (Advantec, Irvine, CA, USA). After obtaining representatives of the different life stages of *Anopheles*, including first, second, third, and fourth instar larvae, as well as pupae and adults, pools of 3 to 5 individuals were formed, along with adults collected using the HLC method (a total of 125 samples), were used for total genomic DNA isolation with the Quick-DNA Tissue/Insect Miniprep Kit (Zymo Research, Irvine, CA, USA). Additionally, samples collected from CDC light traps were processed using the MagMAX™ CORE Nucleic Acid Purification Kit (Thermo Fisher, Waltham, MA, USA).

### 2.4. Molecular Identification of Anopheles Specimens and Endosymbiotic Bacteria

Molecular confirmation of *Anopheles* species was performed using the cytochrome oxidase subunit I gene (COX1) barcode region and PCR conditions previously described [51]. Then, amplicons were bidirectionally sequenced using the Sanger method. Chromatograms were visually checked and edited in MEGAX [52] to obtain the consensus sequences, which were analyzed using the BOLD identification engine tool (http://www.boldsystems.org/). On the other hand, the 16S rDNA, small subunit rDNA, and *Wolbachia* surface protein (*w*sp) gene were used as molecular markers for detection of specific endosymbionts (Appendix A). PCR conditions were followed as described in [53,54,55]. Amplicons were sequenced and submitted to GenBank.

Phylogenetic analyses were performed by downloading 38 COX1 sequences from GenBank. The neighbor-joining method was used to construct the dendrogram for COX1 sequences, the evolutionary model selected was K2P, and a bootstrap value of 1000 was selected. A sequence from *Aedes albopictus* served as an outgroup for the COX1 analysis.

### 2.5. Next-Generation Sequencing of the DNA Samples

A preliminary confirmation of the PCR amplification of the metagenomic DNAs extracted from all samples was initially evaluated using the primer set 27F (5′-AGA GTT TGA TCC TGG CTC AG-3′) and 1492R (5′-GGT TAC CTT GTT ACG ACT T-3′) under the PCR conditions described by other researchers [56] and the resulting amplicons were visualized via agarose gel electrophoresis at 1.2% at 80 V for 45 min. A mock community (Microbial Community Standard by ZYMO) was used as a control to verify the efficiency of the different protocols used for DNA extraction of water and insects.

Fragments and quantities estimations were obtained with a bioanalyzer, and DNAs obtained had sizes greater than 800 bp, on average, in 60% of the fragments detected in all cases in amounts greater than 40 ng of total DNA extracted per sample processed.

Ten ng of those metagenomic DNAs that passed the initial amplification test was further used as templates on PCR amplifications of the V3–V4 hypervariable regions of the 16S ribosomal RNA subunit gene to obtain a PCR product of 470 bp on average, with primers (sequences shown in 5′-3′ direction) 341F: CCTAYGGGRBGCASCAG and 806R: GGACTACNNGGGTATCTAAT [57].

The PCR conditions were performed with concentrations and conditions previously evaluated and reported with Phusion High-Fidelity PCR master mix (New England Biolabs) with 250 picomolar of each primer and including DMSO at a final concentration of 3%. The cycles used were initial denaturation at 98 °C for 2 min, followed by 35 cycles of denaturation 98 °C for 15 s, annealing at 55 °C for 15 s, extension at 68 °C for 30 s, and final step of extension for 5 min. From the amplification products, libraries compatible with Illumina platforms were constructed using the NEBNext Ultra™ II DNA PCR-free Library Prep Kit for Illumina from New England Biolabs and sequenced on the Illumina HiSeq 2500 PE250.

Reads per sample were obtained using 250 cycles on each paired-end direction. The raw paired-end sequence reads dataset are available at the Sequence Read Archive (SRA): SAMN44187221, SAMN44187222, SAMN44187223, SAMN44187224, SAMN44187225, SAMN44187226, SAMN44187227, SAMN44187228, and SAMN44197774.

This 16S rRNA gene sequencing from non-dissected complete anopheline bodies aims to achieve a comprehensive identification of the bacterial community composition at the different stages sampled.

### 2.6. Bioinformatic and Statistical Analysis of the Bacterial Microbiome

In order to determine frequencies of bacterial genera/species in the samples by means of amplicon sequence variants (ASV) classification, we used DADA2 package where filtering by quality, assembly, and chimera cleaning was performed and the taxonomic affiliation of each ASV was defined using RDP classifier, using the consensus of classifications obtained first with Silva 138.1 with assigned species and RDP. Paired datasets per sample were processed using the DADA2 workflow version 1.26.0 and procedures were evaluated to analyze paired reads with parameters considering the number of sequencing cycles and the average amplicon size. The specific parameters used for filtering were the following: out <- filterAndTrim(fnFs, filtFs, fnRs, filtRs, compress = TRUE, truncQ = 2, truncLen = c(226,226), trimLeft = c(1,1), maxN = 0, maxEE = c(2,2), rm.phix = TRUE, matchIDs = TRUE, multithread = TRUE). After taxonomic annotation, curation, and purging of non-relevant data (chloroplasts and animal mitochondria), ASVs still requiring additional curation were analyzed as follows: We proceeded to define whether there is a higher taxonomic resolution for the ASVs found among the 150 samples with the highest total frequency that did not reach a species level annotation with SILVA 138.1 by performing a similarity search against the 16S dataset of the type species reported with an update to the date in LPSN https://lpsn.dsmz.de/ (accessed on 17 August 2024), EzBiocloud, and BlastN searches of the NCBI 16S ribosomal RNA database, Bacteria and Archaea type strains, in case it was possible to more accurately annotate some of these ASVs, and this information was included at species level.

All the ASVs removed were of very low frequency in the samples, indicating that there was amplification that was of high specificity and efficiency for 16S target bacteria and that the most frequent ASVs have a close reference bacterial type species; thus, they may have an accurate taxonomic assignment.

Statistical analysis was conducted using the online tool MicrobiomeAnalyst www.microbiomeanalyst.ca (accessed on 28 September 2024) for graphics of abundances and box-and-whiskers plots of α-diversity considering the Shannon, Chao1, and Simpson indices, and a heat map for representation of the core microbiome. These analyses were performed using analysis of variance (ANOVA) across the entire dataset. The evaluation of β-diversity was conducted using the Bray–Curtis dissimilarity distance and the differences were evaluated by a PERMANOVA. Principal coordinate analysis (PCoA) and non-metric multidimensional scaling (NMDS) plots at the genus level were obtained using the Bray–Curtis index, and correlation networks were obtained using Pearson’s correlation coefficient considering a *p*-value threshold of 0.05. Principal component analysis (PCA) was performed to assess the behavior of physico-chemical parameters of water bodies that were found positive and negative for immature anophelines.

Once relative abundances were obtained, a search for specific endosymbionts was performed to correlate these data with the endosybiont-specific PCR results (Appendix A). A summary of the analyses performed on each sample is presented in Appendix A.

## 3. Results

### 3.1. Characterization of the Anopheles Specimens in the Intradomestic and Peridomestic Areas, Amazonian Secondary Forest, and Breeding Site

Molecular identification through DNA barcoding confirmed the identity of four different *Anopheles* species among 125 specimens collected (Appendix A, Table 1). Two of these species, *An. darlingi* and *An. triannulatus* s.l., are of epidemiological importance.

Notably, *An. darlingi* were collected in four different environments including the intra- and peridomestic areas of S.P.L., with one individual (larva) collected from a flooded canoe (Figure 2C). This canoe was in the same fishing pond where larvae, pupae, and, eventually, adults of *An. triannulatus* s.l. were also found (Figure 2E). Besides the fishing pond, adults of this local malaria vector were also collected in the Amazonian secondary forest (Appendix A). Moreover, in the case of *An. squamifemur,* a search in the literature, as well as in the GenBank, Boldsystems, and Walter Reed Biosystematics Unit (WRBU) databases, revealed that there are no reports of this species in the Amazonas department, this being, to our knowledge, the first time reporting the presence of this species in Amazonian secondary forest from the Amazonas department in Colombia.

The K2P matrix of pairwise distances calculated for the dataset revealed an intraspecific divergence of up to 0.6% in *An. darlingi* samples, while in *An. triannulatus* s.l., distances ranged from 0.31% to 1.27%. Sequences classified as *An. squamifemur* diverged by 0.31% between them.

The phylogenetic tree constructed using the K2P evolutionary model (Figure 3) grouped the *Anopheles* sequences into four main clusters: a first, large cluster comprising sequences from larvae, pupae, and adults of *An. triannulatus* s.l. collected from the fishing pond and Amazonian secondary forest; a second cluster consisting of sequences from a larva collected at the fishing pond and adults of *An. darlingi* from intra- and peridomestic areas; a third cluster containing the single sequence of an adult *An. dunhami* from the Amazonian secondary forest; and a fourth group for *An. squamifemur*, which included both sequences previously classified as such using the Boldsystems database.

### 3.2. Identification of Breeding Sites and Physicochemical Parameters of Water

Due to the extreme high temperatures, the availability of water bodies with potential of being breeding sites was low and of the sites that were found positive with larvae, only one was positive with anophelines. For this reason, results associated with physicochemical properties of water and the bacterial composition associated with this sample are descriptive.

After an active search in the Yahuarcaca watershed at the S.P.L. locality, the positive breeding site of *Anopheles* mosquitoes corresponded to a fishing pond. Aquaculture practices by the community were no longer active, resulting in the absence of fish. Extreme temperatures recorded during the sampling period led to a scarcity of plant debris, while vegetation associated with the water body primarily consisted of grass, low bushes, and aquatic ferns growing along the pond edges. The substrate of the bottom was composed of clay soil, and the presence of microalgae was inferred through visual inspection of the coloration of the stagnant water. The physicochemical water characteristics of this site revealed a COD of 16.00 mg/L, a total iron concentration of 0.59 mg/L, a nitrate concentration of 0.09 mg/L, fixed and volatile solids below 27 mg/L, and a copper concentration below 0.092 mg/L. When comparing to other water bodies in S.P.L. (puddles adjacent to the Yahuarcaca stream), the PCA revealed no differences between the positive breeding site and most of the negative water bodies, with low values of COD, iron, and fixed and volatile solids (Appendix A).

Larvae were collected from two distinct locations within the fishing pond at different times. A single larva, identified as *An. darlingi* (Figure 3), was collected from a flooded canoe inside the pond in November 2023, during extreme dry conditions (Figure 2C). In contrast, 79 larvae and pupae of *An. triannulatus* s.l. were collected directly from the same pond in January 2024, at the onset of the rainy season (Figure 2E), indicating that this fishing pond is potentially a permanent breeding site for anophelines.

### 3.3. Identification of Endosymbiotic Bacteria in Anophelines Collected in S.P.L.-Leticia, Amazonas

Endosymbiont-specific PCR did not yield positive results for any of the endosymbionts searched in the 125 samples processed (Appendix A). False positives were initially detected using the *Arsenophonus* 16S rDNA marker, and subsequently confirmed by Sanger sequencing to correspond to other bacteria, including *Plesiomonas shigelloides*, *Proteus* sp., *Enterobacter* sp., and *Klebsiella* sp. (79.69–100% of identity percentage) in individuals from the fishing pond.

### 3.4. Bacterial Communities and Endosymbiont Presence in Anopheles Larvae, Pupae, Adults, and Breeding Site

After filtering the NGS data, an average of 160,382 counts per sample was obtained, ranging from 41,530 up to 188,851. The rarefaction curve analysis revealed that most samples reached a plateau (Appendix A) and that Good’s coverage for each sample was between 99.99 and 100%, indicating that the sequencing effort captured most of the diversity in the samples. The most abundant bacterial phylum across all samples was Proteobacteria (99.65%), followed by Firmicutes (46.28%) and Bacteroidota (6.89%), across different life stages of *An. triannulatus* s.l. and *An. darlingi* adults. Additionally, besides Proteobacteria, Verrucomicrobiota (53.42%) was the most predominant in the water sample from the breeding site (Figure 4A).

The relative abundances at the genus level showed that *Aeromonas* was the most abundant in *An. triannulatus* s.l. larvae, especially in the third and four instar larvae (57.76%) and adults (52.68%). In the case of the water sample from the breeding site, bacteria from the family Terrimicrobiaceae were the most abundant (51.50%) (Figure 4B). Among the predominant genera, certain bacteria were present across all life stages of *An. triannulatus* s.l. and the water environment, such as *Chromobacterium* (54.03%), *Aquitalea* (4.34%), and *Enterobacter* (48.92%). Other bacteria, such as *Delftia* (15.55%), *Serratia* (32.26%), and *Sphaerotilus* (10.03%), appeared during the early developmental stages, as they were absent from the water sample but present in first and second instar larvae, and persisted throughout later mosquito stages.

It is noteworthy that *Asaia* bacteria were only detected in *An. darlingi* (98.46%), while other low-abundant genera were also present in this species, such as *Delftia* (0.34%), *Chromobacterium* (0.07%), *Enterobacter* (0.03%), and *Serratia* (0.01%), among others. Furthermore, although some genera were not found in all life stages, bacteria like *Klebsiella* and *Enterococcus* were also shared by *An. triannulatus* s.l. and *An. darlingi*.

At the genus level, the core microbiome of the Anopheles species consisted of Aeromonas, Enterobacter, Chromobacterium, Aquitalea, Serratia, Plesiomonas, Paraclostridium, Klebsiella, Sphaerotilus, Clostridium sensu stricto, Enterobacteriaceae, Chryseobacterium, Vogesella, Rhodoferax, Paludibacterium, and Salmonella (Appendix A). None of the endosymbionts targeted by PCR were detected as part of the core microbial community, nor were they present at a low prevalence across the samples, consistent with the negative results from endosymbiont-specific PCR.

### 3.5. Diversity and Microbial Community Structures in the Different Life Stages of An. triannulatus s.l., Its Breeding Site, and An. darlingi Adults

Differences in richness in terms of relative abundance and dominant taxa were similar among samples of *An. triannulatus* (Figure 5). However, the first and second instar larvae of *An. triannulatus* s.l. exhibited a tendency of a higher diversity (Figure 5B,C) when compared to the other samples (Shannon index: 2.31, *p*-value: 0.28, *T*-test statistic: 1.9405; Simpson index: 0.81, *p*-value: 0.42, *T*-test statistic: 1.2518), while overall richness associated with rare or unseen species was significantly different among samples (Figure 5D), with immature individuals of *An. triannulatus* s.l. on the first and second instar being the most diverse (Chao1: 116, *p*-value 0.017, *T*-test statistic: 3.5182; Observed richness was equal to Chao1 index in all cases), followed by individuals of the same species on the third and fourth instar (Chao1:104). In contrast, *An. darlingi* adults showed the lowest bacterial diversity among all samples (Shannon index: 0.10; Simpson index: 0.026; Chao1 index: 19). The results indicated that intraspecific microbial diversity was higher in the early life stages of *An. triannulatus* s.l. larvae with a tendency to decrease in later stages.

Regarding community structure (Figure 5D,E), it was observed that the variable that contributed to significant differences between bacterial communities was species (PcoA: F-value: 2.3931, R-squared: 0.25477, *p*-value: 0.03; and NMDS: F-value: 2.3931, R-squared: 0.25477, *p*-value: 0.028, stress: 0.018854). Specifically, β-diversity analysis revealed distinct differences between bacterial communities at the genus level between immature and adult *An. triannulatus* s.l. samples compared to the bacterial communities in the mosquito breeding site and the adults of *An. darlingi* from the peridomestic area. *An. triannulatus* s.l. groups, regardless of life stage, formed a unique cluster that was significantly different from that of the communities in *An. darlingi* and the fishing pond (Figure 5D), as observed in the principal coordinates analysis (PCoA). Non-metric multidimensional scaling (NMDS) further confirmed significant differences in the bacterial composition between *An. triannulatus* s.l. groups and the rest of the samples (Figure 5E).

The correlation network showed consistency with the community structures and dynamics of the bacteria associated with the core community and the β- diversity representation, describing bacterial community series from the breeding site, followed by larvae, pupae, and adults of *An. triannulatus* s.l., and reflecting a higher diversity in immature stages. The network revealed well-defined clusters of bacteria associated with each life stage of *An. triannulatus* s.l., its breeding site, and the adult of each species (Figure 6), showing significant relationships. The most notable correlations were observed in bacteria from adult mosquitoes, specifically in the *An. darlingi* sample, where *Asaia* exhibited positive correlations with other genera, such as *Staphylococcus*, *Streptococcus*, *Lawsonella*, *Blautia,* and *Methylobacterium* (*r* > 0.93; *p*-value < 0.05 in all cases). In immature stages of *An. triannulatus* s.l. (larvae), *Thorsellia* (Figure 7B) was positively correlated to taxa such as *Azoarcus*, *Klebsiella*, and *Rahnella* (*r* > 0.67; *p*-value < 0.04 in all cases). Additionally, bacteria like *Serratia*, more abundant in pupae and adults (Figure 7A,C), were positively correlated with *Sphingobacterium*, *Tolumonas*, *Duganella*, *Chitinibacter* and *Delftia* (*r >* 0.71; *p*-value < 0.02 in all cases), while *Elizabethkingia* (Figure 7D), more abundant in larvae and adults, was positively correlated to bacteria like *Flavobacterium*, *Flectobacillus*, *Fusibacter*, *Novosphingobium*, and *Sphaerotilus* (*r >* 0.74; *p*-value < 0.02 in all cases). Microbial interactions in the water sample, specifically those with *Bacillus*, though with low abundances (Figure 7E), were positively correlated with genera associated with aquatic environments like the *Terrimicrobiaceae* family (*r =* 0.92; *p*-value: 0.0003), as well as other bacteria, including *Mycobacterium* and *Roseomonas*, some of which are known to pose potential risks to human health.

## 4. Discussion

This study characterized the bacterial composition of two *Anopheles* species found in an indigenous territory from the Colombian Amazon during the dry season [45]. We found that there was a greater bacterial diversity in the larvae and pupae than in the adults of *An. triannulatus* s.l. Additionally, there was a higher bacterial diversity in the adults of *An. triannulatus* s.l. compared to *An. darlingi*. The comparison of the bacterial community composition in the breeding site from which these adults emerged also suggests potential transstadial transmission. This becomes relevant as *An. darlingi* is a major malaria mosquito vector in the Americas [66], while the role of *An. triannulatus* s.l. as a malaria vector is still controversial [67].

In the Amazon rainforest, the migration of both humans and fauna is constantly occurring due to the labor-related issues, deforestation, biodiversity loss, forest resource extraction, and climate change that shape demography and alter vector dynamics, having an impact on the regional epidemiology of the Amazonian countries [68,69,70]. Here, we found four *Anopheles* species in S.P.L. during a dry season, one of them, *An. darlingi,* inhabits intra- and peridomestic environments, as observed in S.P.L., where 26% of *An. darlingi* individuals were collected inside the dwellings. *An. darlingi* is considered a main malaria vector in Colombia with anthropophilic and endophilic behavior and the highest natural infection rates with *Plasmodium* in the Neotropics [71,72,73]. Considering that we also found this species in the peridomestic area and in Amazonian secondary forest, its preference for different environments raises concern about its adaptability in the changing scenarios and migration patterns that constantly occur in the Amazon rainforest and its impact in the spreading of malaria throughout the territory; this finding is in agreement with what has been found by other authors in the Colombian and Brazilian Amazon regarding its anthropophilic and both endophilic and exophagic behavior [73,74].

On the other hand, *An. triannulatus* s.l. is suspected to have a role in malaria transmission [75] as it has been found naturally infected with *P. vivax* and *P. falciparum* [58]. Evidence in Colombia suggests a zoophilic behavior with high abundance in the presence of livestock [62], while there is also evidence of its anthropophilic behavior in the Amazon Basin in Brazil, with mosquitoes of this species living near forest environments, away from the domestic areas [76]. Likewise, none of the *An. triannulatus* collected in S.P.L. were inside dwellings, but in the Amazonian secondary forest and in a fishing pond with no houses nearby.

Regarding *An. dunhami*, our preliminary classification based on classical taxonomy placed it under *An. nuneztovari*, as it was initially considered synonymous with this species, and later with *An. trinkae*, both recognized malaria vectors [64]. Although *An. dunhami* is currently regarded as a non-vector species, as it is not involved in the parasite transmission cycle, its relevance is highlighted by its morphological similarities to other vector species of the Oswaldoi Group [64].

Lastly, *An. squamifemur* collected in the secondary forest of S.P.L. is not a species of epidemiological significance, as there are no reports of its involvement as a malaria vector in the region. Although there is little information available about the ecology of this species, the collection of the mosquito has been conducted inside dwellings as well as outdoors, near the river banks and forests, with zoophilic behavior [77,78,79]. Its presence in the Americas has been recorded in several South American countries [80], with records associated with the Amazon Basin in Venezuela, Bolivia, and Brazil, and in the latter, collected during the high tides of Amazonian rivers [78,79,81], different from this study, where we collected *An. squamifemur* specimens when rivers were at their lowest [45]. In Colombia, it has been recorded in the Llanos Orientales (eastern plains) and in the south Pacific coast [65,77]; however, to our knowledge, this is a new distribution record of the species from the department of Amazonas in the Colombian Amazon.

Overall, larval environments impact adult mosquito fitness [82], with factors such as breeding site type and physicochemical parameters (e.g., salinity, conductivity) influencing the availability, distribution, and abundance of mosquitoes with a tendency of the presence of immature stages of anophelines also associated with total dissolved solids [83], being the latter, along with biochemical oxygen demand and alkalinity, also influencing *Anopheles* abundance [84].

Detoxification mechanisms in anophelines often involve mutations in the voltage-gate sodium channel (*Kdr L1O014F* mutations) and in the Acetilcholinesterase (*Ace-I*) gene [85,86]. In this sense, the physicochemical properties of breeding sites have also been linked to resistance to insecticides such as pyrethroids in anophelines, as evidence suggests that there is a significant correlation between higher frequencies of *Kdr L1O014F* mutations and some water parameters in larval habitats such as salinity, pH, conductivity, and total dissolved solids [87] as well as sulphate, phosphate, potassium, manganese, and iron [88]. In this study, we observed that values of COD, iron, and fixed and volatile solids were low in the fishing pond where immature anophelines were collected, but very similar to those in other sites with no presence of *Anopheles* larvae, indicating that while these parameters play a role in habitat suitability, their individual values may not solely determine the presence of mosquito larvae and further research is needed to explore these complex relationships and identify thresholds of water quality parameters that are possibly involved in resistance mechanisms and that support *Anopheles* development, which could ultimately be considered for effective vector control strategies in aquatic ecosystems.

In addition to physicochemical parameters, microorganisms play a significant role in influencing mosquito larvae, not only serving as a food source but possibly also impacting anopheline physiology and influencing female mosquito behavior by attracting or repelling them to water bodies, thereby affecting oviposition and, consequently, larval distribution [87]. In this study, we observed that total coliforms were present in *An. darlingi*, as well as in both adult and immature stages of *An. triannulatus*, and in the water from the fishing pond. Specifically, *Escherichia* and *Klebsiella* were found in *An. darlingi* and the immature stages of *An. triannulatus*, while *Enterobacter* was detected across all samples. This finding is significant, as fecal coliforms in larval habitats have been shown to influence the presence of other anopheline species, such as *An. coluzzi*, and a non-significant association with higher total coliform levels as observed in *An. gambiae* larvae, suggesting that the first one may prefer aquatic environments where fecal contamination is higher [87]. In contrast, others have found negative correlations with the presence of faecal coliforms and *Anopheles* larvae [89]. Although more research is needed to understand the specific impact of total coliforms on anopheline development, especially in the Amazon, this result highlights the importance of sanitary conditions in Amazonian communities, especially when studies in the Brazilian Amazon suggest that water bodies in peri-urban areas may not fully assimilate anthropogenic pollution on a seasonal scale [90].

Consistent with our findings, other authors [25] reported the presence of *Aeromonas, Bacillus, Klebsiella*, *Enterobacter*, *Stenotrophomonas*, *Aquitalea*, *Serratia*, *Chryseobacterium*, and *Elizabethkingia* in adults of *An. darlingi*, water from their breeding sites, and immature mosquito stages in the Brazilian Amazon. Notably, *Serratia* and *Enterobacter* were found in the water and across all developmental stages, identifying them as potential candidates for paratransgenesis transformation along with *Pantoea*. Ultimately, only *Serratia-*ADU40 and *Pantoea-*Ovo3 proved to be amenable for transformation, with green fluorescent protein expression observed in all life stages, demonstrating their potential for paratransgenesis-based malaria control strategies. In our study, *Serratia* was similarly detected in immature stages and adults of *An. triannulatus* s.l., as well as in *An. darlingi. Pantoea*; however, it was present only in low abundances in *An. darlingi* (0.0039%) and larvae of *An. triannulatus* (0.0052%). Bacterial communities are also influenced by seasonality, and even though we only sampled during the dry season, other authors have reported seasonal differences in microbial richness in *An. coluzzi*, with microbial diversity found to be higher during the dry season in Cameroon and Ghana and no variations observed in families like Enterobacteraceae, with predominant genera like *Enterobacter* and *Pantoea* in mosquitoes collected in the dry season [21,45,91].

Previous studies have reported the presence of endosymbionts such as *Wolbachia* and *Asaia* in laboratory-reared *An. darlingi* specimens, with *Asaia* being detected in low abundance in field-collected mosquitoes [92,93]. Natural infections of *Wolbachia* have been observed in wild specimens of *An. arabiensis*, *An. funestus*, *An. coluzzi*, *An. gambiae*, and *An. moucheti* [31,32,33]. Other endosymbionts, such as *Microsporidia*, have also been detected in *An. gambiae* and *An. coluzzi* in Africa, while *Spiroplasma* has been reported as well in *An. gambiae* [39,40,43]. Here, none of the screened endosymbionts (i.e., *Wolbachia*, *Microsporidia*, *Spiroplasma*, *Cardinium*, or *Arsenophonus*) were detected by either PCR or NGS and further studies are needed, including more species and a wider geographical representation across the range of species distribution to confirm this pattern.

It is worth noting, however, that *Asaia* was present exclusively in *An. darlingi* at a notably high relative abundance in samples collected from the peridomestic area of S.P.L. *Asaia* is a plant-associated bacteria that has exhibited high abundances in sugar-fed mosquitoes [94,95] and has been proposed as a candidate for paratransgenesis, given its ability to transmit horizontally, as observed in adults of *An. gambiae* and *An. stephensi*, as well as its colonization in larvae and newly emerged adults [92]. Additionally, experiments with *Asaia* transformed to express antiplasmodial effectors showed a significant reduction in *P. berghei* oocyst numbers in the gut of *An. gambiae* females [96].

Other symbionts detected in our samples included *Thorsellia*, *Delftia*, *Elizabethkingia*, and *Rosenbergiella*. *Thorsellia*, phylogenetically close to the endosymbiont *Arsenophonus*, was found in larvae and pupae of *An. triannulatus* s.l. and in *An. darlingi*, but not in their breeding site or in adult *An. triannulatus* s.l., which contrasts with the findings in regard to* An. gambiae*, where *Thorsellia* exhibited transstadial transmission from larval habitat to adult stages [97]. *Delftia*, specifically *Delftia tsuruhatensis* TC1, has been studied for its ability to reduce *P. falciparum* numbers in *An. stephensi*, as well as its strong inhibition of oocyst formation for both *P. falciparum* and *P. berghei* in *An. gambiae*, persisting in these mosquitoes from larval to adult stages in laboratory conditions, found in the midgut but not in the ovaries, suggesting it is not vertically transmitted [98]. Although it was not detected in the breeding site, *Delftia* was one of the most abundant genera in our samples, present in both *An. triannulatus* s.l. and *An. darlingi*. Overall, differences between the absence of bacteria in breeding sites and their presence in immature stages and adults suggest the vertical transmission of certain bacterial communities from progenitors, maintained through transstadial transmission, while in adults, the acquisition of some microorganisms may also be linked to their feeding habits in terrestrial environments [27,99]. The discrepancies observed in the microbiota of adults from the two *Anopheles* species could be attributed to this factor, considering that other studies suggest that not only environmental conditions at the breeding site influence the microbiota profile, as meal sources, local weather, and other variables also influence the colonization of bacterial communities in mosquitoes [100,101]. In this study, *An. triannulatus s.l.* adults were collected from the fishing pond, whereas *An. darlingi* adults were collected in the peridomestic area of S.P.L., where they were exposed to different feeding sources. This exposure may have contributed to the distinct microbiota profiles observed in these mosquitoes.

We also detected low abundances of *Elizabethkingia* in *An. darlingi*, as well as in larvae and adults of *An. triannulatus* s.l., and their breeding site. *Elizabethkingia* is relevant as, in mosquitoes like *An. stephensi*, it modulates iron metabolism during blood ingestion, which may play a crucial role in managing oxidative stress caused by excess iron [102]. Lastly, we detected *Rosenbergiella* in low relative abundances in *An. darlingi* (0.030%) and in *An. triannulatus* s.l. larvae (0.0011%). While its role in anophelines remains unclear, in *Ae. albopictus*, it has been shown to prevent viral entry into cells via glucose dehydrogenase action, blocking the transmission of flaviviruses such as dengue and Zika [103]. Microbiota acts as a natural barrier against pathogen infection but can also play a role in the establishment of the *Plasmodium* parasites, and even though we did not search for *Plasmodium* presence in our samples, evidence suggests an increasing *Anopheles* susceptibility to the parasite when treated with antibiotics, while other studies suggest a positive correlation between the presence of *Enterobacteriacea* bacteria and *P. falciparum* [16,104,105], therefore impacting the transmission cycle.

Lastly, limitations of this study were associated with extreme temperatures during the sampling period that limited the availability of suitable collection sites. As a result, the findings related to the breeding site bacterial composition and water sample parameters are descriptive. Additionally, bacterial DNA from *An. darlingi* samples collected in intradomestic areas failed to pass the quality control for NGS, leaving only one sample of *An. darlingi* adults (pool n = 3) available for analysis. Furthermore, due to the scope of the study with a limited number of samples and the absence of recent malaria cases during the sampling period, no tests for *Plasmodium* infection were conducted on the samples, and it is certainly an aspect that would be interesting to explore in future studies at this site under seasonal conditions where more samples could be obtained and analyzed simultaneously with malaria cases being reported in the communities.

## 5. Conclusions

This study found that breeding sites and immature stages of *An. triannulatus* s.l. have a higher bacterial diversity than adults. Furthermore, these data suggest differences in the microbiota associated with *Anopheles* species and correlations among associated bacterial species. Notably, highly abundant bacteria like *Asaia* in *An. darlingi* and *Delftia* in *An. triannulatus* were detected, complying with a relevant feature as potential candidates for developing biological control strategies. Further studies are needed to support these results, including more samples across the distribution ranges of *Anopheles* species and the laboratory characterization of phenotypic effects of those bacterial candidates.

Overall, these findings highlight the significance of understanding mosquito microbiota as a potential tool for malaria control in the Amazon and emphasize the need for further studies to explore the functional roles of these bacteria in mosquito biology and disease transmission.

## Figures and Tables

**Figure 1 insects-16-00269-f001:**
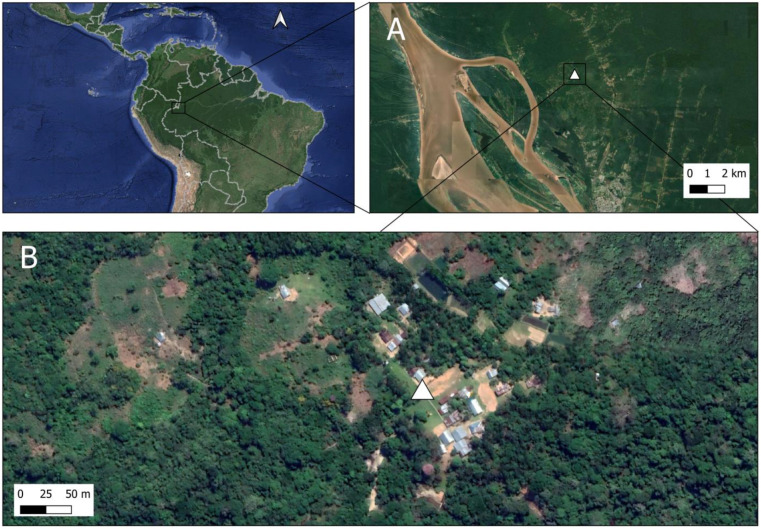
Sample collection area. (**A**) Satellite view of the *Anopheles* collection site in the Amazon basin, Colombia, near the border with the Peruvian and Brazilian Amazon rainforest. (**B**) Indigenous community of S.P.L. and its surrounding area within the Yahuarcaca watershed.

**Figure 2 insects-16-00269-f002:**
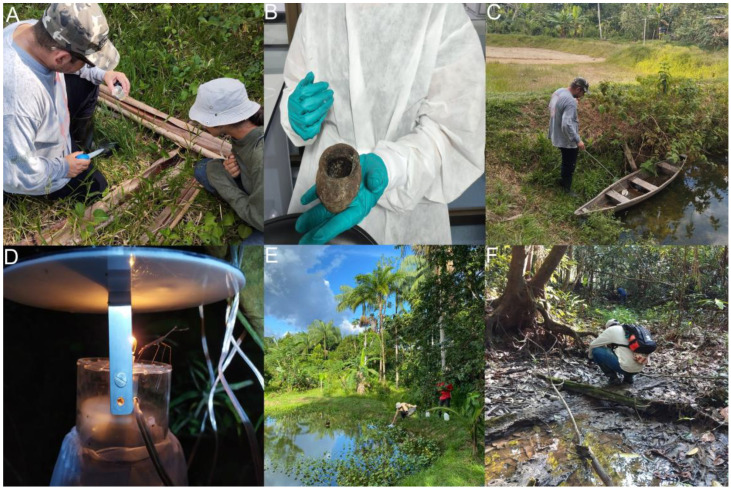
Active search for potential *Anopheles* breeding sites in the Yahuarcaca watershed at S.P.L. locality and installation of light traps. (**A**) Palm tree bracts that store rainwater. (**B**) Seed husk, described by the community as a potential breeding site. (**C**) Abandoned canoe in a fishing pond, flooded by rainwater, without fish inside the pond. (**D**) Collection using CDC light traps before dawn, ensuring 10 h of activity during the night. (**E**) Fishing pond. (**F**) Puddles adjacent to the Yahuarcaca stream.

**Figure 3 insects-16-00269-f003:**
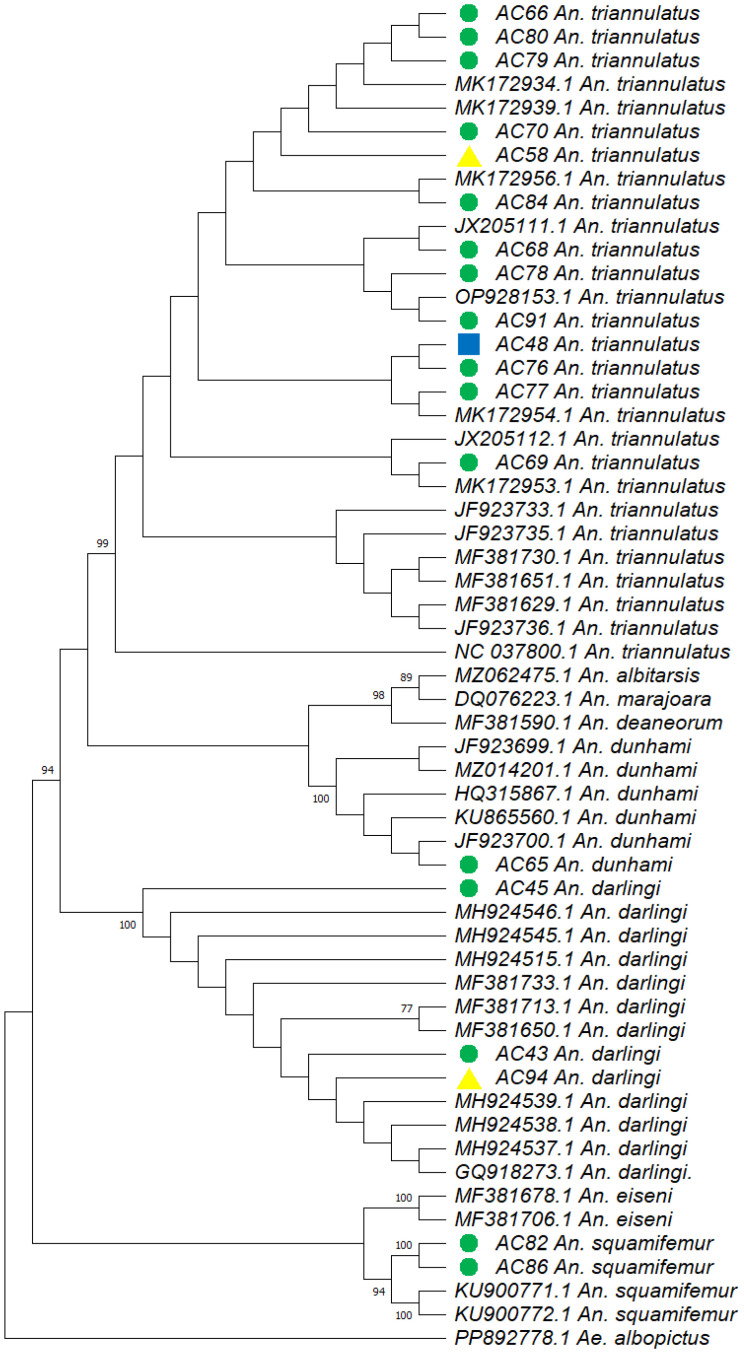
Neighbor-joining phylogenetic tree of the COX1 sequences (710 bp) from anopheline species, reconstructed with the K2P model, and a bootstrap value of 1000. Colors mark the different life stages of the specimens collected in S.P.L., Leticia, Amazonas. Yellow: Larvae. Blue: Pupae. Green: Adults. *Ae. albopictus* was used as an outgroup. Bootstrap values for each cluster are as follows: value of 99 for *An. triannulatus* s.l.; value of 100 for *An. darlingi*; value of 99 for *An. dunhami*; value of 95 for *An. squamifemur*.

**Figure 4 insects-16-00269-f004:**
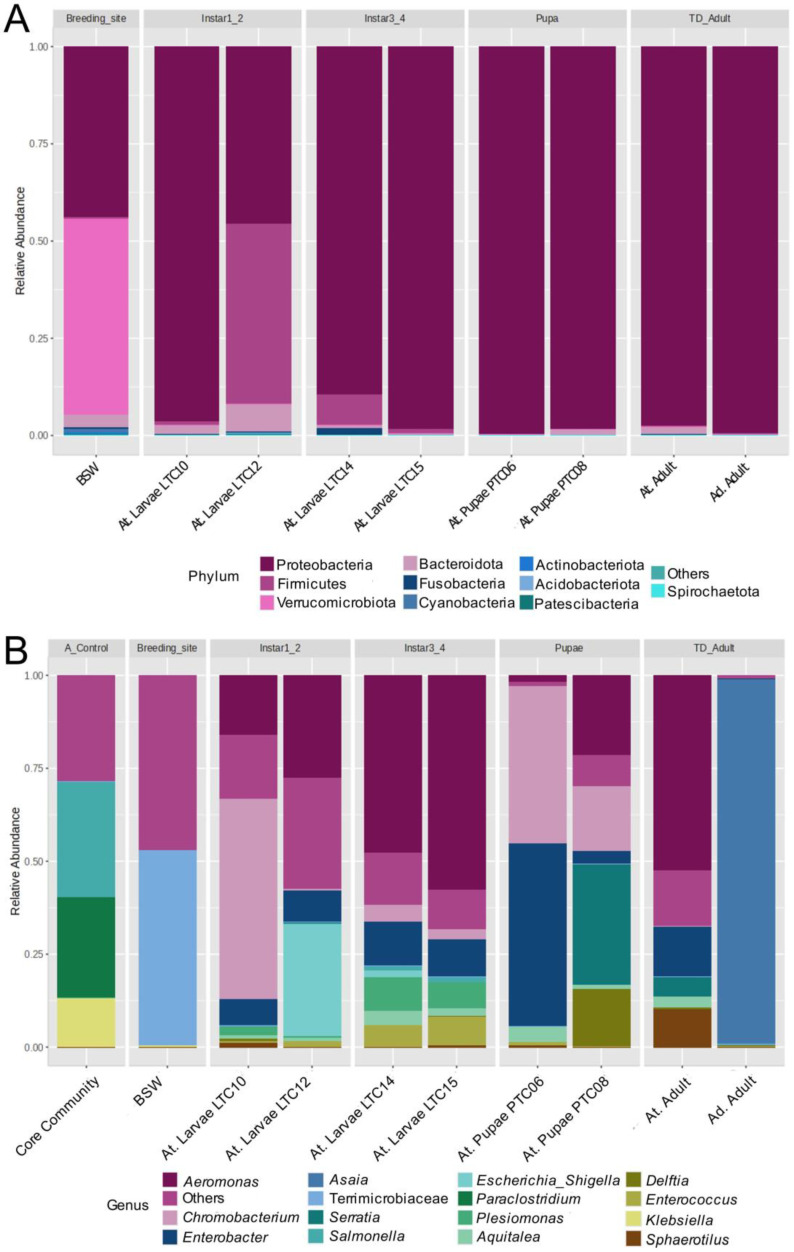
Stacked bar chart of the relative abundance profiling of phyla (**A**) and genera (**B**) found in *An. triannulatus* s.l., its breeding site, and *An. darlingi*. Samples of mosquitoes are grouped by life stage. All samples were collected in the fishing pond, except for the *An. darlingi* pool of adults that were collected from the peridomestic area of S.P.L.-Leticia, Amazonas. The Core Community sample was used as a control. Core Community: Control sample; BSW: Breeding site water; At. Larvae: *An. triannulatus* s.l. larvae samples; At. Pupae: *An. triannulatus* s.l. pupae samples; At. Adult: *An. triannulatus* s.l. adult sample; Ad. Adult: *An. darlingi* adult sample.

**Figure 5 insects-16-00269-f005:**
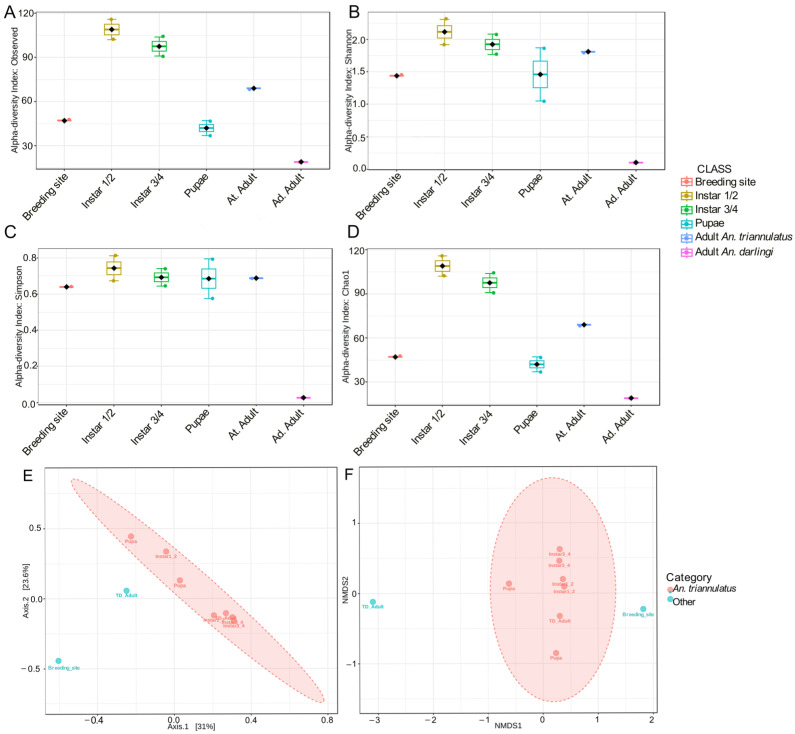
Box-and-whiskers plot representing α-diversity and differences in community structures of immature and adult stages of *An. triannulatus* s.l., adults of *An. darlingi* from the peridomestic area of S.P.L., and *An. triannulatus* s.l. breeding site in terms of (**A**) observed richness (*p*-value 0.017, *T*-test statistic: 3.5182); (**B**) Shannon index (*p*-value: 0.28, *T*-test statistic: 1.9405); (**C**) Simpson index (*p*-value 0.42, *T*-test statistic: 1.2518); (**D**) Chao1 index (*p*-value 0.017, *T*-test statistic: 3.5182); (**E**) PCoA (F-value: 2.3931; R-squared: 0.25477; *p*-value: 0.03); and (**F**) NMDS (F-value: 2.3931; R-squared: 0.25477; *p*-value: 0.028; stress: 0.018854).

**Figure 6 insects-16-00269-f006:**
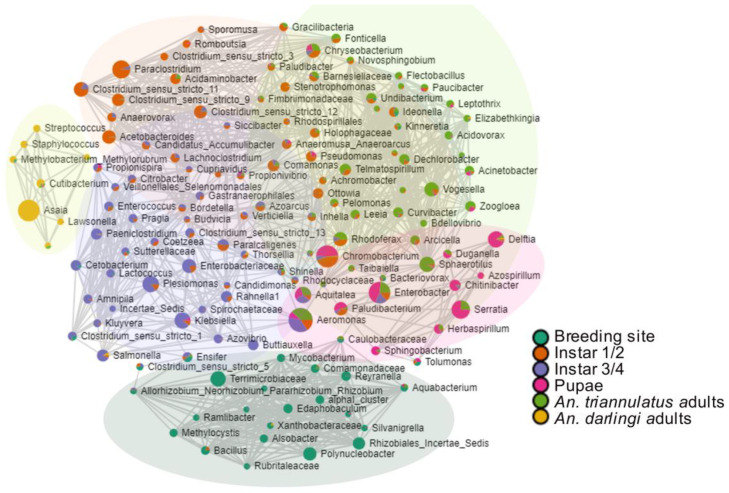
Correlation network using Pearson’s correlation coefficient (*r*) and log-transformed abundances of microbial groups in the different anopheline samples and their breeding site. The green circle represents the cluster associated with bacteria from the water sample. The yellow circle represents the cluster of bacteria from *An. darlingi* adults. Clusters with no circles represent bacteria of immature and adult stages of *An. triannulatus* s.l.

**Figure 7 insects-16-00269-f007:**
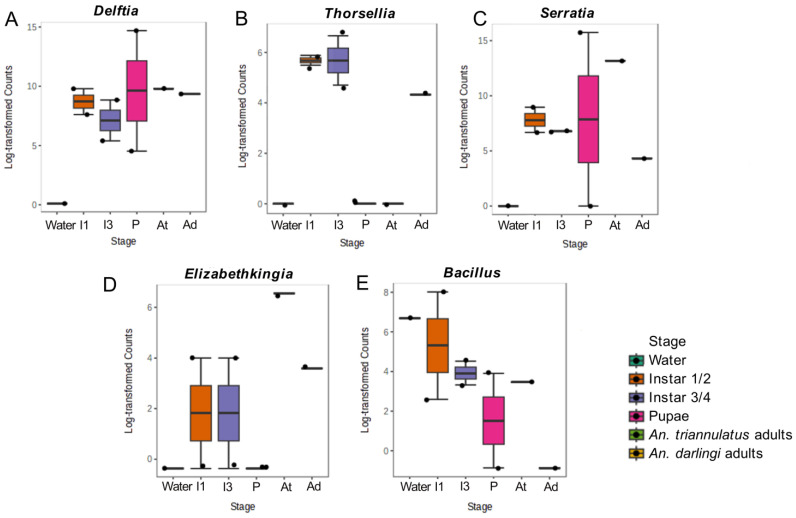
Log-transformed abundances of *Delftia* (**A**), *Thorsellia* (**B**), *Serratia* (**C**), *Elizabethkingia* (**D**), and *Bacillus* (**E**). Water: sample from the breeding site. I1: larvae of *An. triannulatus* s.l. in the first and second instar. I3: larvae of *An. triannulatus* s.l. in the third and fourth instar. P: pupae of *An. triannulatus* s.l. At: adults of *An. triannulatus* s.l. Ad: adults of *An. darlingi*.

**Table 1 insects-16-00269-t001:** Classification by classical taxonomy and molecular identification using the COX1 gene of anopheline specimens collected in S.P.L.-Leticia, Amazonas, Colombia.

Identity by Classical Taxonomy and DNA Barcoding	Life Stage	Number of Individuals	Identity Value (BIN)	Epidemiological Interest	Geographical Distribution in Colombia	References
*An. darlingi*	Larva	1	100% (BOLD: AAA2442)	Main vector. Transmits *P. falciparum, P. vivax,* and *P. malariae*	Caribbean region, Llanos Orientales (eastern plains and grasslands), Pacific coast region (rainforests), Amazon region (rainforest, lowlands, flooded river plains)	[58,59,60]
	Adult	6	100% (BOLD: AAA2442)			
*An. triannulatus* s.l.	Larvae	5	100% (BOLD: AAA9749)	Local vector. Found infected with *P. falciparum* in the Colombian Amazon and *P. vivax* in the Caribbean region.	Caribbean region, Llanos Orientales (eastern plains and grasslands), Amazon region (rainforest)	[58,61,62,63]
	Pupae	5	100% (BOLD: AAA9749)			
	Adult	11	99.39–100% (BOLD: AAA9749)			
*An. dunhami*	Adult	1	99.82% (BOLD: AAA2792)	No	Amazon region (rainforest)	[64]
*An. squamifemur*	Adult	1	99.51% (BOLD: AAJ2766)	No	Pacific coast region, Llanos Orientales (eastern plains and grasslands), Amazon region (rainforest)	[61,65]; Boldsystems database
	Adult	1	99.52% (BOLD: AAJ2766)	No		

## Data Availability

Sequence data associated with the 16S rRNA gene amplicon reads are available in the SRA database under the following codes: SAMN44187221, SAMN44187222, SAMN44187223, SAMN44187224, SAMN44187225, SAMN44187226, SAMN44187227, SAMN44187228, and SAMN44197774. Sequence data associated with the COX1 marker are available in the GenBank database as follows: PQ481937, PQ481938, PQ481939, PQ481940, PQ481941, PQ481942, PQ481943, PQ481944, PQ481945, PQ481946, PQ481947, PQ481948, PQ481949, PQ481950, PQ481951, and PQ481952.

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
