# Peer review of "Exploring the Diversity of Microbial Communities Associated with Two Anopheles Species During Dry Season in an Indigenous Community from the Colombian Amazon"

_insects, 2025, doi:10.3390/insects16030269_

Round 1

Reviewer 1 Report

Comments and Suggestions for Authors

It is a very interesting study due to the search for biological control alternatives as stated by the authors.

In the Material and Methods section, authors should consider giving a more detailed description of the sampling sites or anopheline breeding sites taht there are plant elements that could be important factor for microbial diversity,  such as floating vegetation, submerged vegetation, emergent vegetation, type of substrate at the bottom.

In some parts of the manuscript the way of collecting larvae and pupae is omitted, in addition authors should include the way of preservation of biological material until the moment of appliying techniques.

In the results section, when referring to the correlation, do not use adjetives, it better using the value of correlation so that the reader can have  more accurate view of the association.

If bacteria are not found in mosquito breeding sites, how did the become as endosymbiots?,  requires more details.

The discusión should address the topic of the study, not the role of anophelnea species as vectors, deviate from the objetives and findings of the study.

About 43% of the literatura is recent (no more tan 5 years old).

Author Response

Thank you very much for taking the time to review this manuscript. Please find the detailed responses below and the corresponding revisions/corrections highlighted in the re-submitted files.

2. Point-by-point response to Comments and Suggestions for Authors
Comments 1: Above lines they used the numerical quote form, here they use authors last name and numbers, do not mix, it must be homogeneous  
Response 1: Thank you for pointing this out in line 90. We agree with this comment. Therefore, we have accepted this suggestion and made the corresponding changes in line 90 of the “Corrections” file to adjust the reference accordingly.

Comments 2: How they were collected and the way to preserve the biological material

Response 2: Thank you very much for your suggestion made in line 187. We agree that this part was not properly explained so we added the following paragraph between the lines 213 – 222 of the “Corrections” file: “Larvae and pupae collected from the breeding site were preserved as follows: Large sterile pipettes and larval dippers were used to gather available immature anophelines at the breeding site. Water from the site, along with immature individuals, was placed in a sterile flask covered with a muslin fabric to prevent the escape of newly emerged adults. Larvae were categorized by life stage (instar 1-2 or instar 3-4) and stored in sterile 1.5 mL vials containing ethanol (70%) for molecular analyses. A subset of pupae was immediately stored under the same conditions, while the remaining pupae were allowed to develop into adults to obtain representatives of all life cycle stages (except eggs). All samples were stored at -20°C until DNA extraction was performed (Section 2.3).” 
Comments 3: The line is interrupted

Response 3: Thank you for pointing this out in the line 203. We realized this mistake and corrected the sentence in line 209 of the “Corrections” file: “following the protocol described by other authors”.

Comments/Response 4: Although there is a text box in the line 204, there is no text written in the comment box.

Comments 5: This parameter measures the amount of oxygen needed to oxidize organic matter, it is a parameter to measure the quality of the water. Why the oxygen dissolved in water was not included

Response 5: Thank you very much for taking the time to ask this in the line 205. The reason why we did not measure the dissolved oxygen was because we had a limited number of parameters to request for analysis by the outsourced laboratory, so we unfortunately had to choose what variables were going to be selected, based on the importance of these parameters on the anopheline life cycle, leaving out other parameters that can also be relevant for mosquito development, in this case, dissolved oxygen.

Comments 6: justify why a simulated community

Response 6: Thank you very much for taking the time to ask this in the line 257. The justification of using a mock community was added between lines 270 – 272 of the “Corrections” file: “A mock community (Microbial Community Standard by ZYMO) was used as a control to verify the efficiency of the different protocols used for DNA extraction of water and insects.”

Comments 7: Why in two rows

Response 7: Thank you very much for pointing it out. We accept this suggestion and changed it to one row as observed at the end of Table 1.

Comments 8: Include more about the characterization of the breeding sites, that will give a broader vision of factors that could affect the microbiota

Response 8: Thank you very much for the suggestion made in line 364. We agree that we need to give a more comprehensive characterization of the breeding site, therefore we added this paragraph from lines 393 to 399 of the “Corrections” file, following your suggestion: “Aquaculture practices by the community were no longer active, resulting in the absence of fish. Extreme temperatures recorded during the sampling period led to a scarcity of plant debris, while vegetation associated with the water body primarily consisted of grass, low bushes, and aquatic ferns growing along the pond edges. The substrate of the bottom was composed of clay soil, and the presence of microalgae was inferred through visual inspection of the coloration of the stagnant water.”

Comments 9: A comment on this form of citation was made in another section of the writing

Response 9: Thank you for pointing it out in line 607. We agree with your suggestion and made the changes in the style of referencing accordingly. This change was made in line 640 of the “Corrections” file.

4. Response to Comments on the Quality of English Language

Point 1: In line 103 there is capitalization of the species name.
Response 1: Thank you for pointing this out. We agree with this comment. Therefore, we have accepted this correction and proceeded to change the capitalization of An. atroparvus as suggested in line 104 of the “Corrections” file.

Point 2: In line 124 it is suggested to add “israelensis” to the name Bacillus thurigiensis 
Response 2: Thank you for pointing this out. We agree with this comment. Therefore, we added the word “israelensis” in lines 126/127 of the “Corrections” file.

Point 3: In line 328 there is capitalization of the word “anopheline”

Response 3: Thank you very much for your observation. However, we found out that the word “anopheline” is derived from the latinized genus and used as a regular English word (noun and adjective), so in this case shouldn’t start with an upper-case A.

5. Additional clarifications
Additional comments made directly in the submission page will be responded in this section as follows: 
Comment: In the Material and Methods section, authors should consider giving a more detailed description of the sampling sites or anopheline breeding sites that there are plant elements that could be important factor for microbial diversity,  such as floating vegetation, submerged vegetation, emergent vegetation, type of substrate at the bottom.

Response: Thank you very much for your suggestion. As stated in the previous section in “Response 8” we added this paragraph from lines 393 to 399 of the “Corrections” file, following your suggestion: “Aquaculture practices by the community were no longer active, resulting in the absence of fish. Extreme temperatures recorded during the sampling period led to a scarcity of plant debris, while vegetation associated with the water body primarily consisted of grass, low bushes, and aquatic ferns growing along the pond edges. The substrate of the bottom was composed of clay soil, and the presence of microalgae was inferred through visual inspection of the coloration of the stagnant water.”

Comment: In some parts of the manuscript the way of collecting larvae and pupae is omitted, in addition authors should include the way of preservation of biological material until the moment of appliying techniques.

Response: Thank you very much for your suggestion. As stated in the previous section in “Response 2” we added this paragraph between the lines 213 – 222 of the “Corrections” file: “Larvae and pupae collected from the breeding site were preserved as follows: Large sterile pipettes and larval dippers were used to gather available immature anophelines at the breeding site. Water from the site, along with immature individuals, was placed in a sterile flask covered with a muslin fabric to prevent the escape of newly emerged adults. Larvae were categorized by life stage (instar 1-2 or instar 3-4) and stored in sterile 1.5 mL vials containing ethanol (70%) for molecular analyses. A subset of pupae was immediately stored under the same conditions, while the remaining pupae were allowed to develop into adults to obtain representatives of all life cycle stages (except eggs). All samples were stored at -20°C until DNA extraction was performed (Section 2.3).”

Comment: In the results section, when referring to the correlation, do not use adjetives, it better using the value of correlation so that the reader can have  more accurate view of the association.

Response: Thank you for your suggestion. In the description of results associated with correlation with bacteria we used both adjectives (positive correlation) and their correspondent Pearson’s correlation coefficient in lines 517, 519, 522, 524, and 527 in parentheses of the “Corrections” file.

Comment: If bacteria are not found in mosquito breeding sites, how did the become as endosymbiots?,  requires more details.

Response: Thank you very much for pointing this out. We agree with your suggestion and proceeded to further explain this in lines 688-693 of the “Corrections” file: “Overall, differences between the absence of bacteria in breeding sites and their presence in immature stages and adults suggest the vertical transmission of certain bacterial communities from progenitors, maintained through transstadial transmission, while in adults, the acquisition of some microorganisms may also be linked to their feeding habits in terrestrial environments [27,99].”

Comment: The discusión should address the topic of the study, not the role of anophelnea species as vectors, deviate from the objetives and findings of the study.

Response: Thank you very much for your suggestion. However, we considered this to be important to discuss as, even though the microbiota plays a role in the establishment of some parasites in the mosquito, environmental factors and anthropophilic/zoophilic habits also impact the malaria transmission cycle. In this case, we considered first discussing the habits of the species that we were able to identify and then discussing about microbiota in these insects, approaching the problem from a holistic point of view.

Reviewer 2 Report

Comments and Suggestions for Authors

In the reviewed MS a group of authors from Colombia investigated microbiomes of mosquitos of the genus Anopheles in a single location in Colombia. They screened all collected samples of mosquitos for the presence of symbiotic bacteria (including Wolbachia) via PCR, performed morphological and BLAST identification of the Anopheles spp. form their material, and applied 16s metabarcoding for investigating the bacterial composition of the mosquitos microbiomes. In general the MS is well written and I do not have serious criticism on this MS besides the fact, that the idea of this study is a bit unclear. According to literature and data from 112-121 lines of the MS the generic diversity of symbiotic bacteria of Anopheles spp. is very high. The results of this MS suggest the same. Therefore, it is not clear, how the authors suggest to control the mosquitos if there is no stability in the prokaryote composition? The authors are requested to explain this in the Abstract and in the end of Introduction where they specified the goals of the study.

103 An. Atroparvus – An. a…..

Fig3. Please, root the tree with A.albopictus. Currently it is in the middle of the tree.

Section 3.2. – Please, explain how this section is connected with the main goal of the study.

Author Response

Thank you very much for taking the time to review this manuscript. Please find the detailed responses below and the corresponding revisions/corrections highlighted in the re-submitted files.

3. Point-by-point response to Comments and Suggestions for Authors
Comments 1: In general the MS is well written and I do not have serious criticism on this MS besides the fact, that the idea of this study is a bit unclear. According to literature and data from 112-121 lines of the MS the generic diversity of symbiotic bacteria of Anopheles spp. is very high. The results of this MS suggest the same. Therefore, it is not clear, how the authors suggest to control the mosquitos if there is no stability in the prokaryote composition? The authors are requested to explain this in the Abstract and in the end of Introduction where they specified the goals of the study.
Response 1: Thank you very much for your suggestion. Considering the scarce information about microbiota in anophelines from the Amazon Basin, especially in the Colombian Amazon, the first step is to identify what bacterial groups are part of the mosquito microbiota. For this reason, we agreed with your suggestion and added the following information in the abstract and introduction section of the “Corrections” file in lines 54-57: “Considering the high bacterial diversity observed across the different mosquito life stages, identifying bacterial composition is the first step towards developing new strategies for malaria control. However, the specific roles of these bacteria in anophelines and the malaria transmission cycle remain to be elucidated.” As well as in lines 143-148: “However, the roles of microbiota in anophelines from the Neotropics, particularly the Amazon Basin, and their impact on the malaria transmission cycle remain unclear. Given the limited information on the microbiota composition of these insect vectors in the region, especially in Colombia, it is essential to first identify the bacterial communities and those with biotechnological potential present in these insects.”

Comments 2: Please, root the tree with A.albopictus. Currently it is in the middle of the tree.
Response 2: Thank you very much for pointing it out. We agree with this comment and have updated Figure 3 rooting Ae. albopictus sequence in the tree as suggested. 

Comments 3: Section 3.2. – Please, explain how this section is connected with the main goal of the study.
Response 3: Thank you very much for taking the time to ask this question. Environmental factors have an impact on insect microbiota that can affect the transmission cycle of malaria, and, as seen by studies performed in other insect vectors, as well as in anophelines from Africa, physicochemical parameters of the breeding site water can impact mosquito development as well as bacterial composition in the insect. For this reason, we decided to analyze the water parameters of breeding sites, however characterization of several breeding sites was hindered by the extreme temperatures during the sampling period, and for that, we only characterized the one positive breeding site found. The importance of these physicochemical parameters is discussed as well in lines 600 – 621: “Overall, larval environments impact adult mosquito fitness [82], with factors such as breeding site type and physicochemical parameters (e.g., salinity, conductivity) influencing the availability, distribution, and abundance of mosquitoes  with a tendency of the presence of immature stages of anophelines also associated with total dissolved solids [83], being the latter, along with biochemical oxygen demand and alkalinity, also influencing Anopheles abundance [84]. 
Detoxification mechanisms in anophelines often involve mutations in the voltage-gate sodium channel (Kdr L1O014F mutations) and in the Acetilcholinesterase (Ace-I) gene [85,86]. In this sense, physicochemical properties of breeding sites have also been linked to resistance to insecticides such as pyrethroids in anophelines as evidence suggests that there is a significant correlation between higher frequencies of Kdr L1O014F mutations and some water parameters in larval habitats such as salinity, pH, conductivity, and total dissolved solids [87] as well as sulphate, phosphate, potassium, manganese, and iron [88]. In this study, we observed that values of COD, iron, and fixed and volatile solids were low in the fishing pond where immature anophelines were collected, but very similar to other sites with no presence of Anopheles larvae, indicating that while these parameters play a role in habitat suitability, their individual values may not solely determine the presence of mosquito larvae and further research is needed to explore this complex relationships and identify thresholds of water quality parameters that are possibly involved in resistance mechanisms and that support Anopheles development, which could ultimately be considered for effective vector control strategies in aquatic ecosystems.”

4. Response to Comments on the Quality of English Language
Point 1: In line 103 there is capitalization of the species name.

Response 1: Thank you for pointing this out. We agree with this comment. Therefore, we have accepted this correction and proceeded to change the capitalization of An. atroparvus as suggested in line 104 of the “Corrections” file.

Reviewer 3 Report

Comments and Suggestions for Authors

Duque-Granda et al. focus in the manuscript insects-3354862 on exploring the diversity of microbial communities in two Anopheles species from the Colombian Amazon using 16S rRNA amplicon sequencing. The manuscript is generally well-written; however, I believe the current study is too preliminary and lacks the scientific impact needed to justify publication in Insects. The experimental set-up is poorly replicated with only 9 samples (including a breeding sit water sample), severely limiting the reliability and power of the diversity and network analyses. Furthermore, as the authors acknowledge, there is already a substantial body of research on mosquito microbiota, and this study does not provide any novel insights or significant contributions to the field.

Author Response

Thank you very much for taking the time to review this manuscript. Please find the detailed responses below and the corresponding revisions/corrections highlighted in the re-submitted files.

3. Point-by-point response to Comments and Suggestions for Authors
Comments 1: The manuscript is generally well-written; however, I believe the current study is too preliminary and lacks the scientific impact needed to justify publication in Insects. The experimental set-up is poorly replicated with only 9 samples (including a breeding sit water sample), severely limiting the reliability and power of the diversity and network analyses. Furthermore, as the authors acknowledge, there is already a substantial body of research on mosquito microbiota, and this study does not provide any novel insights or significant contributions to the field.
Response 1: Thank you very much for your feedback. However, we would like you to revise the following justifications as we consider that our research does provide important information about microdiversity of anophelines from the Amazon region and has the scientific accuracy required despite the limitations we encountered during the sampling period:
-    In this study, we processed a total of 125 samples and performed conventional PCR targeting specific markers for identification of 5 bacteria: Wolbachia, Cardinium, Microsporidia, Spiroplasma, and Arsenophonus. Information about these samples can be found in Table S2, however, we also added this in the manuscript in section 2.3 in line 236, and in section 3.3, line 416 of the “Corrections” file. Contrary to what has been reported in Africa, and the few studies that report Wolbachia in specimens from the Amazon region in Brazil, we did not detect any of these bacteria through PCR nor through NGS in the mosquito samples from the Colombian Amazon. However, we were able to detect through NGS the presence of bacteria like Rosenbergiella sp. that has been proposed in Aedes mosquitoes as candidate for biological control, other bacteria such as Delftia, Asaia, and Elizabethkingia that could potentially play a role in the malaria transmission cycle and others such as Klebsiella and Enterobacter that have been studied for their potential use in paratransgenesis, but also are considered bacteria of public health interest.

-    9 samples were indeed sent to NGS, however, these represent 38 individuals categorized into pools from 3 to 5 individuals according to their life cycle (10 larvae 1-2 (2 pools of 5 each), 10 larvae 3-4 (2 pools of 5 each), 10 pupae (2 pools of 5 each), 3 adults of An. triannulatus, 5 adults of An. darlingi, plus the water sample). We realized that this information was not clearly explained and better clarified it in lines 238 – 242 of the “Corrections” file: “A total of 38 individuals, were processed for sequencing using the Illumina Miseq platform (see section 2.5), representing the life stages of An. triannulatus s.l and An. darlingi adults. The specimens were grouped into pools of 5 individuals each, except for three adults that emerged from individuals collected at the fishing pond, which were grouped to form the An. triannulatus s.l adult pool.”

-    - The existing research about mosquito microbiota (anophelines) mainly comes from Africa, in ecosystems that have different dynamics than the Amazon region. Furthermore, the information regarding the Neotropics is scarce, specifically when focusing on Anopheles species that inhabit the Amazon region (where 90% of the malaria cases in South America come from) as the few researches done on anophelines from the Amazon and their microbiota come mainly from Brazil, and a few from Peru. Information about other Amazonic countries (8 in total including those 2 countries) is even more scarce, and to our knowledge, this is the first research done regarding microbiota in anophelines from the Colombian Amazon region. In this regard, novel information is generated from this study, considering that we are also including information about microbiota associated with the life cycle of An. triannulatus s.l. from the Amazon, which is often omitted, and to our knowledge, there is only one other study that analyses this species microbiota collected in the Amazon, but only in larvae (https://journals.plos.org/plosntds/article?id=10.1371/journal.pntd.0007412). Finally, during the survey we also found that there was a new species distribution, An. Squamifemur, that was not previously reported in southern Colombia (Amazonas department) so it is a new record for this Colombian region.

Reviewer 4 Report

Comments and Suggestions for Authors

In this study, Duque-Granda et al. present a detailed characterization of the bacterial microbiota associated with juvenile- and adult-stage anopheline mosquitoes caught in rural habitats within the Amazonas Department of Columbia. This sampling location was thoughtfully chosen as it nears the three-way border intersection of Columbia, Brazil, and Peru, three countries with considerable malaria burdens. Improved knowledge of the microbial communities colonizing anophelines in malaria-endemic countries is highly relevant to epidemiology of this infectious disease, because a burgeoning literature has indicated that the mosquito microbiota significantly modulates both vector competence and propensity to develop insecticide resistance.

Results indicate marked differences in the microbial communities colonizing the two species captured in highest abundance, Anopheles darlingi and An. triannulatus. Several taxa previously reported to colonize mosquitoes were detected in these species, while more in-depth screening for mosquito-specific endosymbionts showed none were present in any samples. Finally,  mosquito sampling from this study resulted in the first report of An. squamifer from southern Columbia. Major strengths of the study include not only the novel information it provides the scientific community but additionally the authors' meticulous attention to detail and rigor regarding methodologies, robust sample sizes, a clear and transparent discussion of limitations of the study, and elegant writing and data presentation.

I have only minor suggestions for further improvement:

- The authors neglected to specify in the Methods section which tissues of mosquitoes (e.g. whole-body, cuticle/carcass, or midgut) were used for total DNA isolation. This is an important point because others have demonstrated that the mosquito external (cuticular) microbiota is not always the same as its internal (gut) microbiota - see for instance Dada et al. 2021 (doi: 10.1186/s12936-021-03934-5). It appears that whole-body insects were sampled without prior surface-sterilization in order to preserve the cuticular microbiota, which is a common approach - can the authors include this in the Methods?

- The text in both the main and supplemental figures is a bit difficult to read due to small font size and low resolution of graphics.

- Several studies suggest that the microbiota of mosquitoes is heavily dictated by the microbial composition of their larval habitat, and that this may be the primary factor driving mosquito microbial community composition while other factors such as mosquito species, habitat type, or sampling season may be of secondary importance (e.g. Coon et al. 2016 (10.1111/mec.13877), Muturi et al. 2018 (doi: 10.1186/s13071-018-3036-9)). The results presented here appear to indicate instead that mosquito species was a stronger driver of microbial composition. I suggest briefly mentioning this interesting discrepancy in the Discussion section.

- Table S1: the reference for Anopheles COX1 primers is highlighted, was this a perhaps a reminder to alter it in some way?

Round 2

Reviewer 3 Report

Comments and Suggestions for Authors

Thank you to the authors for their efforts in revising the article. The microbial community analysis is the core of the paper; however, it lacks effective biological replicates, and its novelty is insufficient. Therefore, I do not recommend its publication in Insects.

Author Response

Thank you very much for taking the time to review this manuscript.